# Sublethal Biochemical Effects of Polyethylene Microplastics and TBBPA in Experimentally Exposed Freshwater Shrimp *Palaemonetes argentinus*

**DOI:** 10.3390/biology12030391

**Published:** 2023-03-01

**Authors:** Juan Manuel Ríos, Andres M. Attademo, Yoshifumi Horie, Paula María Ginevro, Rafael C. Lajmanovich

**Affiliations:** 1Laboratorio de Ecotoxicología, Instituto de Medicina y Biología Experimental de Cuyo (IMBECU), CCT-CONICET, Mendoza 5500, Argentina; 2Laboratorio de Ecotoxicología, Facultad de Bioquímica y Ciencias Biológicas, Universidad Nacional del Litoral (FBCB-UNL-CONICET), Paraje El Pozo s/n, Santa Fe 3000, Argentina; 3Research Center for Inland Seas (KURCIS), Kobe University, Fukaeminami-machi, Higashinada-ku, Kobe 658-0022, Japan

**Keywords:** biomarkers, crustacean, flame retardant, microplastics, plastic additives, toxicity

## Abstract

**Simple Summary:**

This study looked at the effects of exposure to small plastic particles (polyethylene microplastics) and a flame retardant (tetrabromobisphenol A) on the freshwater shrimp *Palaemonetes argentinus*. We used biomarkers such as enzymes and thyroid hormones to assess the sublethal effects after 96 h of exposure. Results showed that the mixture of microplastics and TBBPA at environmentally realistic concentrations led to a decrease in enzyme activities and an increase in T4 hormone levels. These findings suggest that microplastics and plastic additives together could disrupt physiological processes in freshwater crustaceans and ultimately affect upper levels of the food chain.

**Abstract:**

The biochemical effects of sublethal exposure to polyethylene microplastics (PEM) of 40–48 µm particle size and the flame retardant tetrabromobisphenol A (TBBPA), a plastic additive, on the freshwater shrimp *Palaemonetes argentinus* were assessed. Here, we postulate that the use of enzyme and thyroid hormones as biomarkers contributes to the knowledge of the effects of microplastics and plastic additives on freshwater crustaceans. To address this, we evaluated the activities of acetylcholinesterase (AChE), glutathione S-transferase (GST), and carboxilesterase (CbE, using 1-naphthyl acetate (NA) as substrate) and levels of the thyroid hormones thyroxine (T4) and triiodothyronine (T3) after shrimp were exposed (for 96 h) to these xenobiotics at environmentally realistic concentrations. The results showed that the mixture of both xenobiotics led to a decrease in AChE and GST activities and increased T4 levels. We suggest that physiological processes could be compromised in freshwater organisms when exposed to microplastics and TBBPA together, and this could ultimately affect upper levels of the food web.

## 1. Introduction

There is an agreement that plastic waste contamination is one of the global environmental challenges that immediately demands not only methodologies and policies for residue management but also broad knowledge about their real toxic risk for exposed organisms. Hence, environmental research on microplastics (MPs) as emerging contaminants has increased rapidly in the last few years [1]. In nature, combined mechanical, chemical, and microbiological actions can drive the breakdown of large pieces of plastic debris into submillimeter-sized particles, also called MPs (particles < 5 mm), which may be atmospherically transported over a long range to remote areas and even incorporated into freshwater trophic webs [2]. Consequently, the effects of MPs and plastic additives on freshwater organisms are currently the subject of intense experimental research [1]. Numerous up-to date reviews and experimental studies have been published that investigate the environmental problem of MPs contamination and its fate [3,4,5]. To improve our understanding of the fate of MPs and their effects on the biota, it is important to recognize the mechanisms underlying MP uptake by animals. For example, Ma et al. [4], reviewed the effects and fate of MPs after ingestion in several aquatic organisms and proposed that, in general, MP toxicity could be classified as involving the following: (i) physical damage, such as blockage and injury in the digestive tract; (ii) defecation with plastic particles, which disrupts the energy flow of the digestive process; (iii) cause sublethal effects, such as enzyme activity alteration; and (iv) MP accumulation in different tissues and organs.

It has been reported that the intake of these pollutants triggers several consequences, such as intestinal damage in fish and invertebrates [6]. Other adverse effects such as a reduction in food consumption and energy imbalance in fish and crustaceans [7] or facilitation of the allocation of toxic pollutants to the body were also reported [8]. For example, tetrabromobisphenol A (TBBPA), a brominated flame retardant is used as an additive in some plastic products, such as furniture. Some studies reported the presence of TBBPA in freshwater environments [9], and recent evidence suggests that this compound is toxic to the aquatic life belonging to different trophic levels, including freshwater fish [10] and crustacean species [11]. However, knowledge of the biochemical impacts of MPs on aquatic invertebrates is still limited. In crustaceans, for example, it has been reported that exposure to polyethylene MPs causes alterations in the food intake and growth of brine shrimp [12,13] and significantly decreases reproductive performance in copepods [14], brine shrimp [15], and cladocerans such as water fleas [16,17]. Experimental studies conducted with decapod crustacean species highlighted that the most reported effects of MPs are oxidative stress, enzymatic alterations, and reproductive and developmental toxicity [18].

Although there are several metabolic enzymes involved in the homeostasis maintenance of the organisms, the enzymes affected by MP exposure have been insufficiently studied. For example, acetylcholinesterase (AChE) is a key enzyme for nervous system function in both vertebrates and invertebrates [19]. The inhibition of AChE has been widely used as a biomarker to assess the sublethal effects of polyethylene microplastics (PEM) and pesticides in non-target aquatic biota [20]. Carboxylesterase (CbE) isoenzymes catalyze the hydrolysis of a variety of carboxylic esters, including different xenobiotic types [21]. In crustaceans, CbEs play an important role in biochemical processes such as lipid and pesticide metabolism, as reported in the freshwater shrimp *Macrobrachium borellii* [22]. In addition, CbEs catalyze the hydrolysis of plastic additives such as TBBPA in crustacean copepods [11]. Antioxidant enzymes such as glutathione-S-transferases (GST) belong to the phase II detoxification mechanism and provide the first line of tissue defense, and they are often used as xenobiotic biomarkers [23]. The biotransformation of xenobiotics may contribute to the increased production of reactive oxygen species that are potentially toxic for crustacean species such as *Daphnia magna* [24]. On the other hand, it has been reported that thyroid hormones (THs) play a key role in development, metamorphosis, and crustacean metabolism [25]. However, data on the xenobiotic disruption (e.g., by PEM or TBBPA) of thyroid signaling in freshwater crustaceans are unknown so far.

It is important to highlight that the sorption of TBBPA into MPs has been reviewed, indicating severe synergistic effects, e.g., reports include endocrine disorders and reproductive alterations such as proliferation of uterine tumors in female rats (reviewed in Li et al. [26]). Furthermore, Zhang et al. [27] reported that coexposure to MPs and TBBPA together had a greater toxic effect on oxidative stress (antioxidant enzymes: superoxide dismutase, catalase, and glutathione S-transferase) of exposed microalgae than the corresponding single exposure, indicating a synergistic effect of MPs and TBBPA [27].

In this study, we aimed to understand the role of AChE, GST, CbE, and the thyroid hormone (TH: thyroxine (T4) and triiodothyronine (T3)) system in the xenobiotic metabolism when the freshwater shrimp *Palaemonetes argentinus* is exposed to environmental realistic concentrations of PEM and TBBPA, individually and also in combination. To address this, the activities of AChE, GST, and CbE and the levels of T4 and T3 were assessed. We postulate that PEM or TBBPA, as well as the combination of both, evaluated at environmentally realistic concentrations, produce hormonal and enzymatic alterations in exposed shrimp. This information could expand our understanding about the biochemical mechanisms that allow freshwater biota to cope with plastic pollution in their environments.

## 2. Materials and Methods

### 2.1. Test Organism Selection, Collection and Laboratory Maintenance

The freshwater shrimp *P. argentinus* (Crustacea, Decapoda, Palaemonidae) is a ubiquitous species widely distributed in Argentina, Uruguay, Paraguay, and southern Brazil [28]. In Argentina, this shrimp inhabits lagoons, lakes, ponds, dams, rivers, and streams of the La Plata River basin, extending to the provinces of San Luis and Mendoza in the south-west [29]. The freshwater shrimp *P. argentinus* represents a suitable model test organism because: (i) it is sensitive to contaminants of environmental concern [30]; and (ii) its size allows enough tissue to be obtained for biochemical determinations.

*Palaemonetes argentinus* adults (*n* = 50, average wet weight: 0.21 g ± 0.02) used in the present study were caught from ponds associated with the Caroca Stream (33°03′06″ S, 68°56′22″ W) located in the Uco Valley, Central Andes region, Argentina. Shrimp were collected with hand-nets and immediately transported in 30 L containers to our research institution. The collection was made with the permission of the Direction of Renewable Natural Resources (Government of Mendoza, Argentina, research permit #420). Once in the laboratory, shrimp were kept in glass aquariums (330 mm × 170 mm × 205 mm) containing dechlorinated tap water (pH, 7.6 ± 0.1; conductivity, 902 µS cm^−1^; dissolved oxygen, 97%) with constant aeration at 20–22 °C and 11:13 h (light:dark) photoperiod. During the two-week period of acclimation to laboratory conditions before starting the experiment, shrimp were fed daily with a maintenance diet of commercial crustacean pellets (Crusta-Sticks Tropical^®^, T. Ogrodnik, Poland: 30% protein, 3.4% lipids, 3.9% fiber, and 2% phosphorus as shown in the tag). Shrimp were used during the austral early autumn (non-reproductive season) [31].

### 2.2. Experimental Design

To assess possible effects of xenobiotics (PEM and TBBPA) on the activity of AChE, GST, and CbE a-NA, as well as on the concentration of T4 and T3 in shrimp, an in vivo experiment was conducted.

Shrimp were exposed to realistic environmental concentrations of xenobiotic-spiked tap water during 96 h. Shrimp were randomly assigned to the different experimental groups (*n* = 10 each) as follows: Control group, Control-EtOH group, PEM 50 group (polyethylene microplastics, 50 µg L^−1^), TBBPA 5 group (TBBPA, 5 µg L^−1^), and Mix group (a mixture of 50 µg L^−1^ polyethylene microplastics and 5 µg L^−1^ TBBPA). In order to test the effect of TBBPA, absolute EtOH was used to dilute TBBPA correctly and then spiked into the experimental aquarium at the aforementioned nominal concentrations for both the TBBPA 5 group and the Mix group. For this reason, in addition to the Control group (made with dechlorinated tap water), a second control (Control-EtOH) was also used. For possible mortality, shrimp were monitored every 12 h over the 96 h of the whole trial. To prevent both water evaporation from the aquarium as well as cross-contamination with airborne MPs during the experiment, aluminum caps were always used.

Such nominal concentrations were chosen with the aim of imitating an environmentally realistic scenario using the highest concentrations reported in water samples for PEM [32] and TBBPA in freshwater environments [33]. For example, Rodrigues et al. [24] reported a high peak concentration of PEM in river water samples that reached 51 µg L^−1^. On the other hand, Liu et al. [33] reported that the most serious case of TBBPA pollution in China was found in lake water samples with concentrations of TBBPA reaching 5 µgL^−1^. Therefore, in this study we wanted to emulate a maximum exposure scenario that can occur under natural conditions. Polyethylene microplastics were purchased from Sigma-Aldrich (40–48 µm particle size; density 0.9215–166 0.9255 gm L^−1^; purity > 99%; CAS number 9002-88-4, Saint Louis, MO, USA). We chose this particle size because it is within the size range of the items that make up the natural diet of this species of freshwater shrimp. The only study on natural diet of wild *P. argentinus* [29] reported that the juveniles and adults of this freshwater shrimp species feed on Euglenophyceae algae, Bacillariophyceae, filamentous algae, plant remains, Rotifera, Nematoda, Copepoda, Cladocera, Oligochaeta, Chironomidae larvae, and Insecta larvae. All these prey items (or fragments) have a size range broader than the size of the MPs used (40–48 µm particle size) in the present experiment. The standard TBBPA (97% pure grade, CAS number 79-94-7, lot#MKCM2562) was purchased from Aldrich Chemical Company (Saint Louis, MO, USA).

After the 96 h exposure experiment, shrimp were sacrificed on ice for 10 min, weighed and kept cold until preparation of the homogenate with the shrimp whole body as follows: for enzyme kinetic assays, *P. argentinus* was weighed (g) and homogenized (1:10, *w*/*v*) in cold 25 mM sucrose, 20 mM Tris-HCl buffer (pH = 7.4) containing 1 mM EDTA, using a glass-PTFE Potter-Elvehjem tissue grinder connected to a Heidolph type ST1. The homogenates were centrifuged at 10,000× *g* for 15 min at 4 ± 1 °C and kept at −80 °C until biochemical analysis. The Biuret method was used to determine protein concentrations to determine enzymatic activities [34].

### 2.3. Biomarker Determinations

The activity of acetylcholinesterase (AChE, EC 3.1.1.7), glutathione S-transferase (GST, EC 25.18.1), and carboxylesterase (CbE, EC 3.1.1.1) was determined as an indicator of xenobiotics exposure. AChE activity was determined according to Ellman et al. [35]. The reaction mixture (final volume = 930 µL) consisted of 25 mM Tris-HCl buffer containing 1 mM CaCl_2_ (pH = 7.6), 10 µL 20 mM acetylthiocholine iodide, 50 µL 300 µM 5,5′-dithiobis-2-nitrobenzoic acid, and 20 µL sample. Variation in optical density was determined from duplicate samples at 410 nm and 25 °C for 1 min using a Jenway 6405 UV–VIS spectrophotometer. Enzyme activity was expressed as nmol min^−1^ mg^−1^ of protein, using a molar extinction coefficient of 13.6 × 10^3^ M^−1^ cm^−1^. All enzymatic activities assessed here were measured at 25 °C (Jenway 6405 UV-VIS spectrophotometer).

GST was determined spectrophotometrically using the method described by Habig et al. [36] and modified by Habdous et al. [37]. The enzyme assay was performed at 340 nm in 100 mM Na-phosphate buffer (pH 6.5) (F.V. = 920 µL), 20 µL of 0.2 mM 1-chloro-2,4-dinitrobezene, 50 µL of 5 mM reduced gluthatione, and the sample. Enzyme kinetics assays were performed at 25 °C and whole GST activity was expressed as nmol min^−1^ mg^−1^ of protein, using a molar extinction coefficient of 9.6 × 10^3^ M^−1^ cm^−1^.

CbE activity was determined using 1-naphthyl acetate (1-NA) as a substrate. The hydrolysis of 1-NA was determined according to Gomori [38] and adapted by Bunyan and Jennings [39]. The reaction medium (1940 µL) consisted of 25 mM Tris-HCl, 1 mM CaCl_2_ (pH = 7.6), and 10 µL of the supernatant (sample). After a 5 min pre-incubation period, the reaction was initiated by adding 50 µL of 1-NA (46 µM, in acetone) and incubated at 25 °C for 10 min. The formation rate of naphthol was stopped by adding 500 µL of 2.5% (*w*/*v*) SDS and subsequently 0.1% (*w*/*v*) of Fast Red ITR dissolved in 2.5% (*w*/*v*) Triton X-100. The samples were left in the dark for 30 min for color development. The absorbance of the naphthol–Fast Red ITR complex was read at 530 nm (using a molar extinction coefficient of 33.225 × 10^3^ M^–1^ cm^–1^).

The levels of T4 and T3 were measured using enzyme-linked electro-chemiluminescent immunoassay (ECLIA) kits (COBAS^®^, Roche Diagnostics, Indianapolis, IN, USA) following the protocol previously described in Attademo et al. [40]. The detection limits for T3 and T4 were 0.0001 ng g^−1^ and 2.1 ng g^−1^, respectively.

### 2.4. Statistical Analyses

Non-parametric statistics were used since the raw data did not fit a normal distribution (Shapiro–Wilks W test and Kolmogorov–Smirnov test). Therefore, to test for differences among study experimental groups, the Kruskal–Wallis (KW) *H* test, followed by a posteriori multiple comparisons of mean ranks for all groups, was used. Statistical analysis was carried out using the InfoStat 2008 software [41]. A *p* value < 0.05 was considered significant.

## 3. Results

No mortality was observed during the entire 96 h experiment. Significant statistical differences were found among experimental groups when comparing the median AChE (KW *H* = 15.20, *p* = 0.004) and GST activity (KW *H* = 12.80, *p* = 0.012) in shrimp. Post-hoc analysis revealed that such differences were due to a high inhibition of AChE activity (48% and 43% compared with Control and Control-EtOH, respectively) and GST activity (43% and 40% compared with control and Control-EtOH, respectively) of exposed shrimp belonging to the Mix group (Figure 1A). Post-hoc analysis also revealed that statistically comparable AChE and GST activities were found among the Control, Control-EtOH, PEM 50, and TBBPA experimental groups (Figure 1B). No significant statistical differences were found among experimental groups when comparing the median CbE a-NA activity (KW *H* = 3.20, *p* = 0.524) of shrimp (Figure 1C).

Regarding TH levels, a significant statistical difference was found among experimental groups when comparing the median T4 levels (KW *H* = 10.91, *p* = 0.027) in the whole shrimp body. Post-hoc analysis revealed that such a difference was due to an increase in the T4 level of shrimp belonging to the Mix group (Figure 2), while comparable levels of T4 were found among the remaining experimental groups (Figure 2). No significant difference (KW *H* = 8.78, *p* = 0.066) was found among experimental groups when comparing the T3 level in shrimp (Figure 2). Here, the overall results obtained from the xenobiotic exposure experiment showed that the mixture of PEM and TBBPA at environmentally realistic levels led to a significant decrease in AChE and GST activities and increased T4 levels in exposed shrimp. These finding are discussed below.

## 4. Discussion

The mechanisms underlying the activities of AChE, CbEs, and GST in decapod crustacean species exposed to MPs have been little explored so far. To date, there is only one study that has evaluated the effect of exposure to MPs on AChE and GST activity in the crab *Charybdis japonica* [42], while the activity of GST after exposure to MPs has been evaluated for the shrimp *Litopenaeus vannamei* [43,44]. Regarding AChE results from the present study, we found a significant inhibition of AChE activity only by the PEM and TBBPA mixture, suggesting that the single action of each compound was not sufficient to modify AChE activity. In this regard, Picó et al. [45] reported that different types of compounds with similar mechanisms of action could cause a synergistic effect. However, these results highlight the need to study not only simple binary mixtures, but also multi-compound cocktails for a comprehensive estimation of the effects of co-occurring pollutants.

As regards GST results from the present study, we found a significant inhibition of GST activity after coexposure to the PEM and TBBPA mixture. This suggests that: (i) the single action of each compound was not sufficient to alter GST activity; and (ii) GST participates in detoxification, and physiological processes could be compromised in freshwater shrimp coexposed to PEM and TBBPA. These findings were surprising because GST plays a key role in the detoxification of xenobiotics and may contribute to defending tissues from oxidative stress by increasing its activity [46]. GST activity is commonly used as an indicator of alteration of phase II of the biotransformation by the antioxidant defense system [23]. Specifically, GST catalyzes the conjugation of glutathione with a diversity of xenobiotics, thereby neutralizing its active electrophilic sites and later making the conjugated compound more hydrophilic, as reported for other crustacean species such as *Daphnia magna* [24]. Recent studies conducted with decapod species have reported a significant induction in GST activity after microplastics exposure [42,43,44], contrasting with the results of this study. The absence of an increment in GST activity in response to xenobiotic exposure used in our experiment may be linked to inactivation of the enzyme by toxicants or to a reduction in glutathione conjugation, leading GST to deplete its activity [47], or to a physiological adaptation of these organisms to the pollutants from the experimental groups. Previous studies have reported single effects of other emergent contaminants on GST levels of freshwater biota after acute experiments. For example, *Rhinella arenarum* tadpoles treated with glyphosate herbicide for 48 h showed an inhibition of GST activity [48]. More studies on the response of this enzyme are needed to determine the differences between the effects of acute versus chronic exposure (e.g., an exposure greater than 96 h).

To our knowledge, there is only one study reporting specific CbE activity in adult decapods. Chen et al. [49] reported an inhibitory effect on CbE activity (using acetyl-CoA as substrate) in adults of the redclaw crayfish *Cherax quadricarinatus* exposed to polystyrene microspheres. In the present study, the activity of CbE (using 1-NA as substrate) in *P. argentinus* was not affected by any of the treatments carried out. Since there are no studies for adult decapods on specific CbE activity using 1-NA substrate hydrolysis, it is not feasible to establish comparisons with the present study.

Although disruption of the thyroid hormones (THs) has been reported in zebrafish exposed to TBBPA alone [50], MPs [51], and the combination of MPs with plastic additives [52], there are no reports on the effect of MPs or plastic additives on the regulation of THs in crustaceans to date. The neuroendocrine cells of decapod crustaceans located in hormone secretion tissues can secrete numerous hormones, the release of which into the hemolymph plays key regulatory roles in the ontogenetic development of organisms [53,54]. In decapod crustaceans, the thyroid hormones thyroxine and triiodothyronine play a pivotal role in regulating metabolism and metamorphosis [25]. Recently, Dvoretsky et al. [25] reported that both thyroid hormones were detected in the red king crab *Paralithodes camtschaticus* hemolymph. The authors’ findings confirm that TH levels changed significantly depending on the age of crabs and sampling season, and therefore are involved in the specific physiological mechanisms [25].

Recently, Han et al. [55] reported that the coexposure of microplastics (polystyrene) and bisphenol A retard gonadal development of the whiteleg shrimp *Litopenaeus vannamei* by disrupting gonad-inhibiting hormone (GIH) and molt-inhibiting hormone (MIH) regulation in an in vivo experiment. Authors found that both xenobiotics together were more toxic than the corresponding single exposures, which may be triggered by the Trojan horse effect and summation of the toxic impacts on common targets [55]. Our research supports these former assumptions since we found that shrimp coexposed to PEM and TBBPA showed an increase in the levels of the hormone T4. This finding would suggest that both compounds could act concomitantly on the regulation of T4 in shrimp, which could presumably be due to a Trojan horse effect. The mechanistic basis of this hypothesis, however, deserves additional analysis. On the other hand, as regards the T3 level results, no differences were detected among experimental groups. Presumably, the threshold of the alterations caused by the xenobiotics on T3 levels in exposed shrimp here, could ostensibly be lower than that for T4 levels, and so no differences were detected. However, additional research is still needed to understand the impact of MPs and plastic additives on the regulation of TH levels to address this issue with more certainty.

## 5. Conclusions

Shrimp upregulated their detoxification capacity through their antioxidant defense (via GST) in response to the exposure of both xenobiotics combined (PEM and TBBPA together). However, this mixture also reduced neural activity (reduced AChE). Results suggest that freshwater crustaceans have an inherent capacity to counter the acute effects of both microplastics and plastic additives, but there is a limit beyond which the defense mechanisms fail and therefore physiological functions are compromised. As plastic contamination will deteriorate in the future, plastic waste may undermine the performance of freshwater organisms, affecting upper levels of the food web and thus the structure and functionality of freshwater ecosystems. The present study offers valuable evidence on the impact of plastic waste pollution on freshwater biota and suggests that the biomarkers evaluated here are useful tools in environmental risk assessments of these emerging pollutants.

## Figures and Tables

**Figure 1 biology-12-00391-f001:**
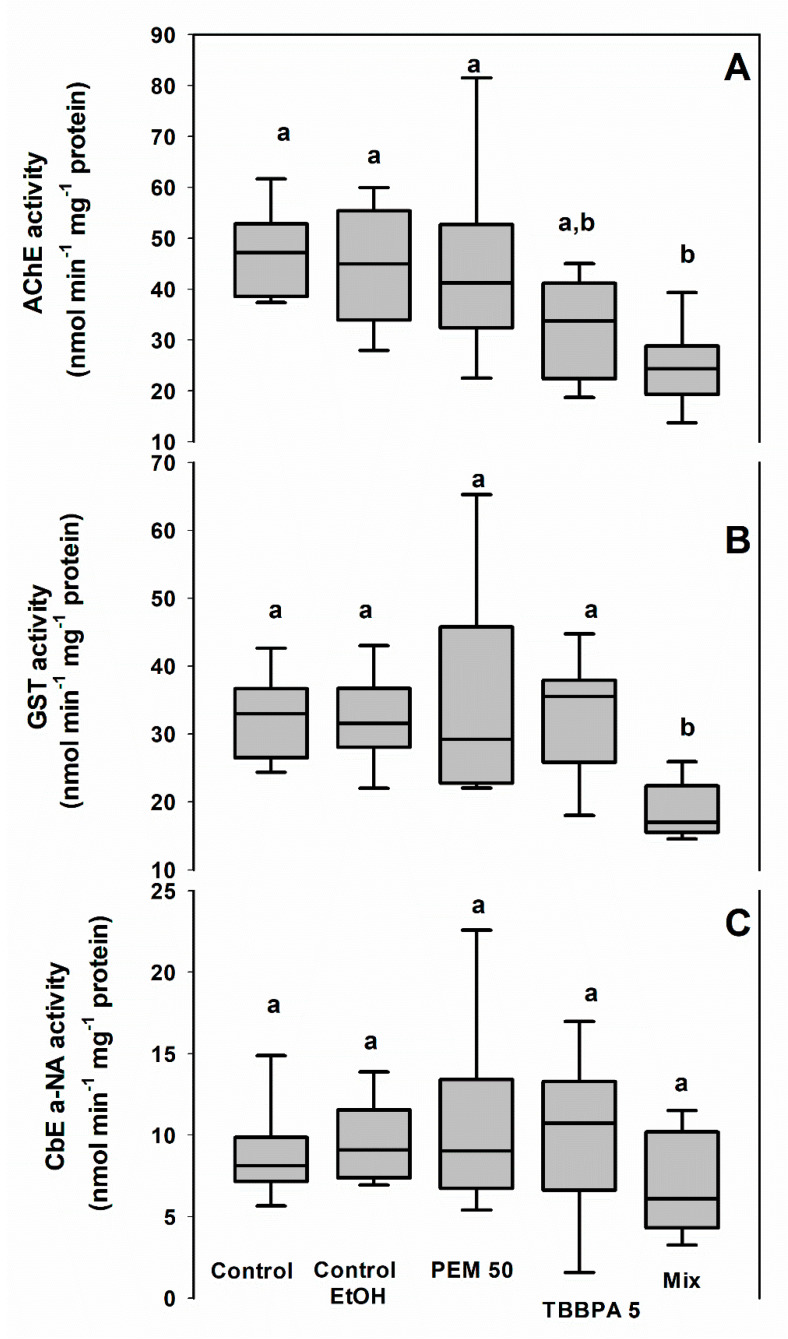
Variation of (**A**) acetylcholinesterase (AChE), (**B**) glutathione S-transferase (GST), and (**C**) carboxylesterase 1-naphthyl acetate (1-NA) activities in shrimp whole body after 96 h of exposure to the following different experimental groups (*n* = 10 for each aquarium): Control, Control-EtOH, PEM 50 (polyethylene microplastics, 50 µg L^−1^), TBBPA 5 (TBBPA, 5 µg L^−1^), and Mix group (mixture of 50 µg L^−1^ polyethylene microplastics and 5 µg L^−1^ TBBPA). Tukey box plots indicate the median, the 25th and 75th percentiles (box edges), and the range (whiskers). Different lowercase letters denote significant differences among experimental groups (*p* < 0.05, Kruskal–Wallis *H* test, followed by a posteriori multiple comparisons of mean ranks for all groups).

**Figure 2 biology-12-00391-f002:**
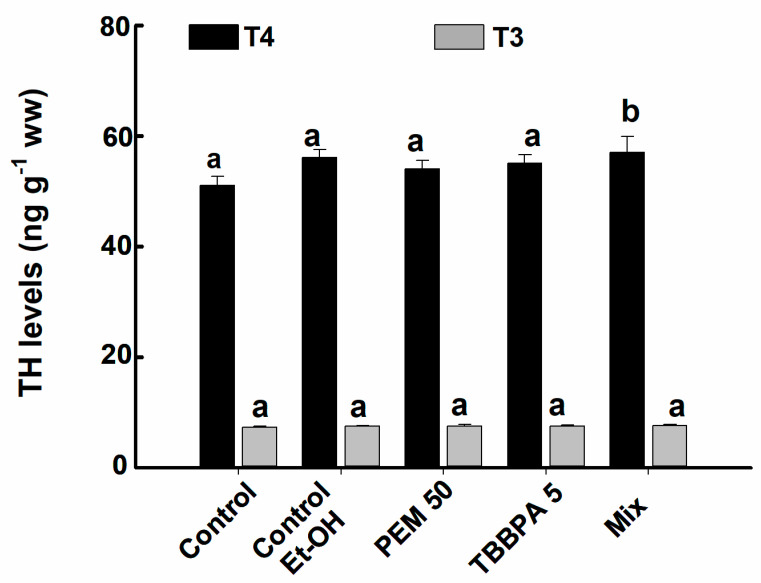
Level of thyroxine (T4) and triiodothyronine (T3) in shrimp whole body after 96 h of exposure to the following different experimental groups (*n* = 10 for each aquarium): Control, Control-EtOH, PEM 50 (polyethylene microplastics, 50 µg L^−1^), TBBPA 5 (TBBPA, 5 µg L^−1^), and Mix group (the mixture of 50 µg L^−1^ polyethylene microplastics and 5 µg L^−1^ TBBPA). Data are expressed as median ± SE. Different lowercase letters denote significant differences among experimental groups (*p* < 0.05, Kruskal–Wallis *H* test, followed by a posteriori multiple comparisons of mean ranks for all groups).

## Data Availability

All data generated or analyzed during this study are included in this manuscript.

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
