# Peer review of "Sublethal Biochemical Effects of Polyethylene Microplastics and TBBPA in Experimentally Exposed Freshwater Shrimp *Palaemonetes argentinus"

_biology, 2023, doi:10.3390/biology12030391_

Round 1
Reviewer 1 Report
This work explored the biochemical sublethal effects of microplastics and TBBPA on the freshwater shrimp Palaemonetes argentinus. The authors found that the mixture of both xenobiotics led to a significant decrease in AChE and GST activities, and increased T4 levels. This work showed that physiological processes could be compromised in freshwater organisms when exposed to microplastics and TBBPA together, and could ultimately affect upper levels of the food web. The manuscript is in general well written and the method development process is described in detail. However, the following issues have to be addressed before this manuscript is suitable for publication.
1. In the section Introduction, paragraphs 1-2, more details about the endocrine disrupting activities of microplastics and plastic additives (including TBBPA), as well as their joint toxicity should be provided.
2. The section Results and discussion should be divided into 2-3 parts.
Author Response
Mendoza, February 22
Response Letter
Dear Ms. Shelley Wang, Dr. Silvia Galafassi and Dr. Pietro Volta
Guest Editors – BIOLOGY. Special Issue entitled "Micro and Nanoplastics in Freshwater Fauna: Sources, Quantification and Effects”
Thanks a lot for your response and attention to our manuscript (Ms. 2174310), Title: "Biochemical sublethal effects of polyethylene microplastics and TBBPA
in experimentally exposed freshwater shrimp Palaemonetes argentinus".
Appended to this letter you will find our detailed responses to all comments and suggestions, including modifications to the section: Introduction, Results and discussion, Conclusions, and References list. Besides, as per the suggestion of the journal management, Brief Report has been changed to Article.
Please note: Reviewers’ comments are written as regular text, while our responses are denoted by bold text, and italicized text denotes modified portions of the manuscript transcribed to the present response letter. Besides, we track our changes in the manuscript by using blue in the word document. We hope this revised version of the manuscript is suitable for publication in Water. We thank you for considering it and we look forward to reading from you soon.
Kind regerds,
Dr. Juan Manuel Ríos
----------------------------------------------------------------------------------------
Reviewer 1
This work explored the biochemical sublethal effects of microplastics and TBBPA on the freshwater shrimp Palaemonetes argentinus. The authors found that the mixture of both xenobiotics led to a significant decrease in AChE and GST activities, and increased T4 levels. This work showed that physiological processes could be compromised in freshwater organisms when exposed to microplastics and TBBPA together, and could ultimately affect upper levels of the food web. The manuscript is in general well written and the method development process is described in detail. However, the following issues have to be addressed before this manuscript is suitable for publication.
- In the section Introduction, paragraphs 1-2, more details about the endocrine disrupting activities of microplastics and plastic additives (including TBBPA), as well as their joint toxicity should be provided.
Au: To enrich the Introduction we include a brief paragraph (new paragraph 3) with the information requested by Reviewer 1 as follows:
It is important to highlight that the sorption of TBBPA into MPs has been reviewed, indicating severe synergistic effects such as endocrine disorders and reproductive toxicity, and may increase the incidence of uterine tumors in female rats (reviewed in Li et al. [26]. Furthermore, Zhang et al. [27] found that co-exposure to MP and TBBPA together had a greater toxic effect on oxidative stress (antioxidant enzymes: superoxide dismutase, catalase, and glutathione S-transferase) of exposed microalgae than the corresponding single exposure, indicating an evident synergistic effect of MPs and TBBPA [27].
- The section Results and discussion should be divided into 2-3 parts.
Au: Amended. Results and discussion have been divided in this new version as requested.
---------------------------------------------------------------------------------------

Reviewer 2 Report
The manuscript of Juan Manuel Ríos and colleagues is focused on a well-investigated topic in the last years, which is rich of interesting literature on regards. Despite this, some important aspects of the effect of microplastic pollution on living organisms are unknown, especially the ones related to the synergic effects with other pollutants normally present in aquatic environments. I found this information the most important of this document, however is affected in my opinion by some issue to address.
The English language needs a general revision to enrich it, at the present form sounds rather clear but also basic in several parts of the document.
The main weakness of this study that I found is that there is not an experimental evidence of MPs ingestion by the model organisms. This is essential in my opinion to evaluate the different effects of the concentrations tested and properly discuss the results of the entire study relating to the conclusions. Can the authors solve this?
Keywords contains some words already reported in the Title. This is always suggested to avoid, using different words to increase the soundness of the document.
The first period of the introduction needs the support of more references related to the general MPs' pollution fate, which today is rich of interesting experimental and review manuscripts. See for example:
https://doi.org/10.1016/j.scitotenv.2020.142572
https://doi.org/10.1016/j.envpol.2020.114089
https://doi.org/10.1016/j.scitotenv.2020.139436
https://doi.org/10.1016/j.jhazmat.2020.124187
"In crustaceans, for example, it has been re- ported that MPs exposure significantly decreased reproductive output in copepods [9] the brine shrimp Artemia franciscana [10] and Daphnia magna [11]". Not only on this but important effects were also reported for biological essential functions as feeding behavior and growth. Please enrich this period to give more soundness to your manuscript using more related references, see for example:
https://doi.org/10.1016/j.envpol.2018.10.024
https://doi.org/10.3390/app11083352
https://doi.org/10.1016/j.envpol.2019.113233
In the Material and Methods section should be well exposed and argue the MPs particles dimension chosen, to relate it on the diet of the model organism. Is Palaemonetes argentinus common to feed on particles of this dimension in nature, and in the area you picked them up from?
Moreover, you stated to have sampled your studied organisms from some "unpollutted ponds", on which base you stated this? You have carried out analysis to support the environmental pollution of the area?
Table 1 reports unnecessary information, because of rather obvious in a short term exposure experiments, in my opinion could be avoided to give more fluency to the manuscript.
"However, these results highlight the need to study not only simple binary mixtures, but also more complex combinations for a comprehensive estimation of the effects of co-occurring pollutants." Despite this assumption is true, it is so utopian because it is almost impossible to reproduce in the laboratory the mixture of pollutants and other chemical ions that interact simultaneously in the natural environment, and with organisms.
"The absence of an increment in GST activity in response to xenobiotic exposure used in our experiment, may be related to inactivation of the enzyme by toxicants, or to depletion of glutathione conjugation, leading GST to lose its activity [39], or to a physiological adaptation of these organisms to the pollutants from the experimental groups." This can also depend on the size of the particles and whether the animals were able to ingest it or not, have you verified this thing and have images in support?
No discussion about the effect on the T3 hormone, why the authors think it is particularly irrelevant?
Please double-check the references style and italicize the scientific names.
Best regards
The Reviewer
Author Response
Mendoza, February 22
Response Letter
Dear Ms. Shelley Wang, Dr. Silvia Galafassi and Dr. Pietro Volta
Guest Editors – BIOLOGY. Special Issue entitled "Micro and Nanoplastics in Freshwater Fauna: Sources, Quantification and Effects”
Thanks a lot for your response and attention to our manuscript (Ms. 2174310), Title: "Biochemical sublethal effects of polyethylene microplastics and TBBPA
in experimentally exposed freshwater shrimp Palaemonetes argentinus".
Appended to this letter you will find our detailed responses to all comments and suggestions, including modifications to the section: Introduction, Results and discussion, Conclusions, and References list. Besides, as per the suggestion of the journal management, Brief Report has been changed to Article.
Please note: Reviewers’ comments are written as regular text, while our responses are denoted by bold text, and italicized text denotes modified portions of the manuscript transcribed to the present response letter. Besides, we track our changes in the manuscript by using blue in the word document. We hope this revised version of the manuscript is suitable for publication in Water. We thank you for considering it and we look forward to reading from you soon.
Kind regards,
Dr. Juan Manuel Ríos
----------------------------------------------------------------------------------------
Reviewer 2
The manuscript of Juan Manuel Ríos and colleagues is focused on a well-investigated topic in the last years, which is rich of interesting literature on regards. Despite this, some important aspects of the effect of microplastic pollution on living organisms are unknown, especially the ones related to the synergic effects with other pollutants normally present in aquatic environments. I found this information the most important of this document, however is affected in my opinion by some issue to address.
The English language needs a general revision to enrich it, at the present form sounds rather clear but also basic in several parts of the document.
Au: Since R1 version of MS was revised by the official translator of our institute, the English language was enhanced along the entire manuscript as requested.
The main weakness of this study that I found is that there is not an experimental evidence of MPs ingestion by the model organisms. This is essential in my opinion to evaluate the different effects of the concentrations tested and properly discuss the results of the entire study relating to the conclusions. Can the authors solve this?
Au: In the present study, we did not assessed MPs ingestion by shrimps during the bioassay. At the end of the experiment, every shrimp whole body was used to make the homogenates to evaluate the biomarkers (enzymes and hormones). We decided to do this because i) enough homogenate was needed to assess the target biomarkers, ii) since they are small animals and we used an acceptable number to carry out the statistical analyses, we could not use some individuals to analyze their digestive tracts through staining studies with nile red and fluorescence microscopy.
Therefore, currently, we aren´t able to know if the biochemical effects observed were due to ingestion of polyethylene particles (with TBBPA) by the shrimps or were due to the contact with the gills during filtration and diffusion through countercurrent flow. However, prior to perform the bioassay, we had supposed about this point and that is why it is planned, shortly, to carry out a second study to elucidate MPs intake by shrimp. This will be done by putting the MPs in the water (one group), by putting the same MP in commercial crustacean food (second group) plus two controls. The second experimental instance that includes food with MPs, will be carried out in order to elucidate changes in the morphology of the gut but also the activity of digestive enzymes.
Keywords contains some words already reported in the Title. This is always suggested to avoid, using different words to increase the soundness of the document.
Au: Some keywords were changed as requested:Keywords: biomarkers; crustacean; flame retardant; microplastics; plastic additives; toxicity
The first period of the introduction needs the support of more references related to the general MPs' pollution fate, which today is rich of interesting experimental and review manuscripts. See for example: https://doi.org/10.1016/j.scitotenv.2020.142572
https://doi.org/10.1016/j.envpol.2020.114089
https://doi.org/10.1016/j.scitotenv.2020.139436
https://doi.org/10.1016/j.jhazmat.2020.124187
Au: We really appreciate the reviewer's suggestion. The first period of the introduction has been improved, using the suggested bibliography, as follows: Numerous up-to date reviews and experimental studies have been published that invest the environmental problem of MPs contamination and its fate [3, 4, 5]. To improve our understanding on the fate of MPs and their impact on the biota, it is important to understand the mechanisms underlying MPs uptake by animals. For example, Ma et al. [4], reviewed the effects and fate of MPs after ingestion in several aquatic organism, and proposed that, in general, MPs toxicity could be classified as follows: i) accumulation within the digestive tract, causing physical damage such as clogging and injury; ii) discharge as pseudofeces, which disturbs the energy flow of organisms; iii) cause sublethal effects such as enzyme activity alteration; and iv) translocation within the body, which exposes the internal tissues and organs to MPs.
"In crustaceans, for example, it has been reported that MPs exposure significantly decreased reproductive output in copepods [9] the brine shrimp Artemia franciscana [10] and Daphnia magna [11]". Not only on this but important effects were also reported for biological essential functions as feeding behavior and growth. Please enrich this period to give more soundness to your manuscript using more related references, see for example:
https://doi.org/10.1016/j.envpol.2018.10.024
https://doi.org/10.3390/app11083352
https://doi.org/10.1016/j.envpol.2019.113233
Au: We truly appreciate this suggestion. The text was enhanced using the reviewer 2 suggested bibliography. The paragraph was enriched as follows: In crustaceans, for example, it has been reported that exposure to polyethylene MPs cause alterations in the feeding behavior and growth of brine shrimp [12; 13], and significantly decreased reproductive performance in copepods [14], brine shrimp [15] and cladocerans such as water fleas [16, 17].
In the Material and Methods section should be well exposed and argue the MPs particles dimension chosen, to relate it on the diet of the model organism. Is Palaemonetes argentinus common to feed on particles of this dimension in nature, and in the area you picked them up from?
Au: To clarify, we include the following sentence in M&M section as follows:
We chose this particle size because it is within the size range of the items that make up the natural diet of this species of freshwater shrimp. The only study on natural diet of wild P. argentinus [29] reported that the jueveniles and adults of this freshwater shrimp species feed on algae Euglenophyceae, Bacillariophyceae, filamentous algae, plant remains, Rotifera, Nematoda, Copepoda, Cladocera, Oligochaeta, Chironomidae larvae, and Insecta larvae. All these prey items (or fragments) have a size range broader than the size of the MPs used (40-48 µm particle size) in the present experiment.
(21) Collins, P. A. 1999. Feeding of Palaemonetes argentinus (Decapoda: Palaemonidae) from an oxbow lake of the Paraná River, Argentina. Journal of Crustacean Biology, 19(3), 485-492.
Moreover, you stated to have sampled your studied organisms from some "unpollutted ponds", on which base you stated this? You have carried out analysis to support the environmental pollution of the area?
Au: In the M&M section we claimed that “Palaemonetes argentinus adults (n = 50, average wet weight: 0.21g ± 0.02) used in the present study were obtained from unpolluted ponds associated to Caroca Stream…” Since such ponds are currently used for the breeding of sogyo fish (Ctenopharyngodon idella) for sport fishing purposes, the owners regularly carry out water quality controls to ensure that the fishing preserve is free of pesticides and heavy metals Besides, we only got permission from the owner to catch the shrimp, but no water samples for chemical analysis. Nevertheless, we have decided to remove the word "unpolluted" to avoid confusion for readers.
Table 1 reports unnecessary information, because of rather obvious in a short term exposure experiments, in my opinion could be avoided to give more fluency to the manuscript.
Au: Amended. To give more fluency to the manuscript, Table 1 was removed as suggested by Reviewer 2.
"However, these results highlight the need to study not only simple binary mixtures, but also more complex combinations for a comprehensive estimation of the effects of co-occurring pollutants." Despite this assumption is true, it is so utopian because it is almost impossible to reproduce in the laboratory the mixture of pollutants and other chemical ions that interact simultaneously in the natural environment, and with organisms.
Au: We agree with reviewer 2 assertion. It is utopian to carry out the perfect experiment indeed.
"The absence of an increment in GST activity in response to xenobiotic exposure used in our experiment, may be related to inactivation of the enzyme by toxicants, or to depletion of glutathione conjugation, leading GST to lose its activity [39], or to a physiological adaptation of these organisms to the pollutants from the experimental groups." This can also depend on the size of the particles and whether the animals were able to ingest it or not, have you verified this thing and have images in support?
Au: We did not verified particles in shrimp gut nor have images of intestinal tracts after the 96hs trial. See above our answer to reviewer 2 regarding this same point and our justification of using the shrimp whole body homogenates preparation to have enough sample to biochemical measurements.
No discussion about the effect on the T3 hormone, why the authors think it is particularly irrelevant?
Au: To clarify, a brief sentence regarding this point was included at the end of the paragraph in the discussion section as follows:
On the other hand, as regards the T3 levels results, no differences were detected among experimental groups. Presumably, the threshold of the alterations caused by the xenobiotics for T3 levels in exposed shrimps here, could ostensibly be lower than that for T4 levels and so no differences were detected. However, additional studies are still needed to understand the effects of MPs and plastic additives on the regulation of THs levels to address this issue with even modest certainty.
Please double-check the references style and italicize the scientific names.
Au: Amended. References style were double-checked and the scientific names were italicized.

Reviewer 3 Report
1. I recommend authors not to use abbreviations where possible. Why should the reader memorize the abbreviations used in the text?
2. “This shrimps, integrates the diet of several groups of aquatic animals such as fish, amphibians and birds”. Edit this sentence, please.
Author Response
Mendoza, February 22
Response Letter
Dear Ms. Shelley Wang, Dr. Silvia Galafassi and Dr. Pietro Volta
Guest Editors – BIOLOGY. Special Issue entitled "Micro and Nanoplastics in Freshwater Fauna: Sources, Quantification and Effects”
Thanks a lot for your response and attention to our manuscript (Ms. 2174310), Title: "Biochemical sublethal effects of polyethylene microplastics and TBBPA
in experimentally exposed freshwater shrimp Palaemonetes argentinus".
Appended to this letter you will find our detailed responses to all comments and suggestions, including modifications to the section: Introduction, Results and discussion, Conclusions, and References list. Besides, as per the suggestion of the journal management, Brief Report has been changed to Article.
Please note: Reviewers’ comments are written as regular text, while our responses are denoted by bold text, and italicized text denotes modified portions of the manuscript transcribed to the present response letter. Besides, we track our changes in the manuscript by using blue in the word document. We hope this revised version of the manuscript is suitable for publication in Water. We thank you for considering it and we look forward to reading from you soon.
Kind regards,
Dr. Juan Manuel Ríos
---------------------------------------------------------------------------------------
Reviewer 3
I recommend authors not to use abbreviations where possible. Why should the reader memorize the abbreviations used in the text?
Au: Amended. Several abbreviations were replaced by the extended names in some paragraphs along the entire manuscript.
“This shrimps, integrates the diet of several groups of aquatic animals such as fish, amphibians and birds”. Edit this sentence, please.
Au: We have decided to remove this sentence, which has no impact on the manuscript.

Round 2
Reviewer 2 Report
Dear Authors,
I appreciated your accuracy in reviewing the manuscript, despite some points being unsolvable because of the experimental design. Further manuscripts are needed to clarify some aspects not considered in this one, for the future, consider including all these data together to enhance the value of your future studies.
Best regards
The Reviewer
Author Response
Dear Reviewer 2
Thank you very much for your quick response. Yes, in the future we will take your suggestions into account